# *USP15-USP7* Axis and *UBE2T* Differential Expression May Predict Pathogenesis and Poor Prognosis in De Novo Myelodysplastic Neoplasm

**DOI:** 10.3390/ijms241210058

**Published:** 2023-06-13

**Authors:** Luiz Gustavo Almeida de Carvalho, Tatiana Takahasi Komoto, Daniel Antunes Moreno, João Vitor Caetano Goes, Roberta Taiane Germano de Oliveira, Mayara Magna de Lima Melo, Mariela Estefany Gislene Vera Roa, Paola Gyuliane Gonçalves, Carlos Victor Montefusco-Pereira, Ronald Feitosa Pinheiro, Howard Lopes Ribeiro Junior

**Affiliations:** 1Center for Research and Drug Development (NPDM), Federal University of Ceara, Fortaleza 60020-181, CE, Brazil; lgcarvalho.biomed@hotmail.com (L.G.A.d.C.); vitorc.biomed@gmail.com (J.V.C.G.); maymagna21@gmail.com (M.M.d.L.M.); cmontefusco@gmail.com (C.V.M.-P.); pinheirofeitosa@gmail.com (R.F.P.); 2Post-Graduate Program in Translational Medicine, Federal University of Ceara, Fortaleza 60020-181, CE, Brazil; 3Molecular Oncology Research Center, Barretos Cancer Hospital, Barretos 14784-390, SP, Brazil; tattytk@hotmail.com (T.T.K.); daniellmoreno@gmail.com (D.A.M.); paolagyulliane@gmail.com (P.G.G.); 4Post-Graduate Program of Pathology, Federal University of Ceara, Fortaleza 60020-181, CE, Brazil; 5Post-Graduate Program in Medical Science, Federal University of Ceara, Fortaleza 60020-181, CE, Brazil; 6Post-Graduate Program of Immunology, University of São Paulo, São Paulo 14040-902, SP, Brazil; mariela.biomed@gmail.com; 7Department of Pathology, School of Medicine, Universidade Estadual Paulista, Botucatu 18618-970, SP, Brazil

**Keywords:** ubiquitination, deubiquitination, gene expression, myelodysplastic neoplasm

## Abstract

The aim of this study was to evaluate the expression of *USP7*, *USP15*, *UBE2O*, and *UBE2T* genes in Myelodysplastic neoplasm (MDS) to identify possible targets of ubiquitination and deubiquitination in MDS pathobiology. To achieve this, eight datasets from the Gene Expression Omnibus (GEO) database were integrated, and the expression relationship of these genes was analyzed in 1092 MDS patients and healthy controls. Our results showed that *UBE2O*, *UBE2T*, and *USP7* were upregulated in MDS patients compared with healthy individuals, but only in mononucleated cells collected from bone marrow samples (*p* < 0.001). In contrast, only the *USP15* gene showed a downregulated expression compared with healthy individuals (*p* = 0.03). Additionally, the upregulation of *UBE2T* expression was identified in MDS patients with chromosomal abnormalities compared with patients with normal karyotypes (*p* = 0.0321), and the downregulation of *UBE2T* expression was associated with MDS hypoplastic patients (*p* = 0.033). Finally, the *USP7* and *USP15* genes were strongly correlated with MDS (r = 0.82; r2 = 0.67; *p* < 0.0001). These findings suggest that the differential expression of the *USP15-USP7* axis and *UBE2T* may play an important role in controlling genomic instability and the chromosomal abnormalities that are a striking characteristic of MDS.

## 1. Introduction

Myelodysplastic neoplasm (MDS) is a heterogeneous group of clonal hematological disorders characterized by impaired bone marrow (BM), ineffective hematopoiesis causing chronic and progressive peripheral cytopenia, cellular dysplasia, and functional abnormalities of blood cells, in addition to an increased risk of progression to Acute Myeloid Leukemia (AML) [1,2]. MDS is predominantly an elderly disease, although it can occur at any age. About 75% of patients with MDS are over 60 years of age when diagnosed, and the incidence rate doubles every decade after age 40 [3]. In Brazil, the average age at diagnosis is 68 years [4].

Aging is the most important risk factor for the development of MDS. Due to errors in DNA replication and spontaneous mutations, its accumulation in hematopoietic stem cells generates cell survival advantage, thus giving rise to clonal hematopoiesis. Examples of specific mutations include the *RUNX1*, *TP53*, and *SF3B1* genes [2,5]. Additionally, mutations in genes involved in epigenetic regulation, such as *DNMT3a* and *TET2*, are associated with MDS [2,5]. About 80% of patients diagnosed with MDS have at least one somatic mutation, and half of the patients may have chromosomal alterations [2].

The ubiquitin-proteasome system (UPS) is an important non-lysosomal proteolytic system that plays a crucial role in the regulation of eukaryotic cell functions. It acts as a component of a catabolic pathway that regulates the intracellular breakdown of proteins, and as a non-proteolytic pathway that regulates the location and activity of several cellular proteins, including cell cycle, apoptosis, differentiation, DNA repair, histone modification, and other biological/epigenetic functions [6]. Ubiquitin is the founding member of a family of structurally conserved proteins that regulate several processes in eukaryotic cells by exerting functions through covalent binding to other cellular proteins and, therefore, altering protein stability, localization, and target activity. Protein substrates are recognized and degraded by the catalytic component now known as the 26S proteasome [7,8].

Ubiquitination is a post-translational modification that regulates various cellular processes, mainly protein catabolism, and it plays a crucial role in the development of many diseases, such as cancer [9]. The cellular functions of ubiquitination comprise a range of proteolytic and non-proteolytic processes, such as protein degradation by proteasomal pathways, intracellular transit, inflammatory signaling, autophagy, DNA repair, and regulation of enzyme activity. Therefore, its deregulation can lead to multiple consequences, such as the activation or deactivation of important pathways involved in cell metabolism and even pathways related to oncogenes, which can be harmful to cell homeostasis [8].

Ubiquitin-conjugating enzyme E2T (UBE2T), also known as HSPC150, was initially identified as an important element in the Fanconi anemia (FA) pathway. UBE2T participates in FANCD2 monoubiquitination, a fundamental process for DNA damage repair and maintaining chromosomal stability [10]. In addition, the ubiquitin-conjugating enzyme E2O (UBE2O) is a hybrid enzyme that has both E2 and E3 activities, and it can ubiquitinate proteins independently of E3 [6,11]. It also repairs DNA as an inhibitor of double-stranded breaks (DSBs) mediated by homologous recombination (HR); thus, it is involved in the regulation of aging and the onset of cancer [12]. This justify both genes *UBE2T* and *UBE2O*, and their respective protein expressions, being associated with several cancers and diseases, such as hepatocellular carcinoma [13], multiple myeloma (MM) [14], bladder [15], gastric cancer [16], multiple myeloma [11], breast and prostate cancer [6,17], and mixed-lineage leukemia [18].

Deubiquitinating enzymes (DUBs) belong to the isopeptidase family and play the opposite role of ubiquitination [6]. There is increasing recognition of DUBs that are mutated in human cancers, suggesting their role as oncogenes and tumor suppressors [19]. Human ubiquitin-specific protease 7 (USP7) and ubiquitin-specific protease 15 (USP15) both regulate several biological processes and tumorigenesis, including DNA damage response, transcription, epigenetic control of gene expression, immune response, cell proliferation, apoptosis, autophagy, cell cycle, cell invasion, tumor genome integrity, and transcription regulation [20,21]. Given the involvement of USP7 in multiple cellular pathways, its expression is often dysregulated in human malignancies, making it an oncogene that promotes tumor growth and negatively affects the immune response to tumors [22], such as in chronic lymphocytic leukemia, acute lymphocytic leukemia (ALL), and multiple myeloma [23,24]. In short, USP7 plays an important role in a variety of pathologies and is a good target for therapeutic and inhibitor development [22].

On the other hand, *USP15* may act as an oncogene and a tumor suppressor in different contexts, including being an attractive therapeutic target [21]. DUBs are essential in preserving normal hematopoiesis. *USP15* is highly expressed in human hematopoietic tissues and leukemias, and it is reported as a critical DUB in protecting genome integrity in hematopoietic stem cells and leukemia cells, playing an important role in the preservation of all major hematopoietic differentiation pathways [25].

The role of *USP7*, *USP15*, *UBE2O*, and *UBE2T* gene expression in MDS is unclear. Thus, this report aims to validate the expression of these genes to identify possible targets of ubiquitination and deubiquitination in the pathobiology of de novo MDS in a Brazilian cohort.

## 2. Results

### 2.1. Prediction In-Silico Analysis Demontrated That UBE2O, UBE2T, and USP7 Are Upregulated in Caucasian MDS Patients

The differential gene expression analysis for *UBE2O*, *UBE2T*, *USP7*, and *USP15* was based on microarray data from 1092 patients previously published by our research group [26]. We compared gene expression data from patients with MDS, obtained from primary mononuclear bone marrow (PMBM) cells or CD34+ isolated stem cells, to healthy controls [26]. The analysis of the mononuclear BM cells microarray identified that *UBE2T* (MDS mean: 7.021 versus Control mean: 5.625; *p* < 0.001), *UBE2O* (MDS mean: 8.693 versus Control mean: 7.461; *p* < 0.001), and *USP7* (MDS mean: 12.245 versus Control mean: 11.620; *p* < 0.001) were all upregulated in PMBM cells of MDS patients compared with healthy controls (Figure 1). However, *USP15* did not show any expression during this analysis.

On the other hand, none of the evaluated genes showed differential expression in the microarray data analyzed for CD34+ stem cell samples from patients with MDS (Figure 2).

### 2.2. Validation of UBE2O, UBE2T, USP7, and USP15 Gene Expression in Brazilian MDS Patients

#### 2.2.1. Characteristics of MDS Patients

The features of the MDS patients are presented in Table 1. We validated the in-silico data on the expression of *UBE2O*, *UBE2T*, *USP7*, and *USP15* in samples of mononuclear BM cells in a Brazilian cohort comprising 72 MDS patients and four bone marrow samples from sex- and age-matched healthy individuals. All healthy individuals showed a normal karyotype.

MDS patients were clinically classified based on the new WHO classification launched in 2022 (WHO 2022), using morphological criteria. Thus, MDS patients were categorized as follows: 24 (33.3%) patients with MDS with low blasts (MDS-LB), 17 (23.6%) with MDS with low blasts and ring sideroblasts (MDS-RS-LB), and 3 (4.2%) patients with MDS hypoplastic (MDS-h). As for patients with increased blasts, 18 (25.0%) patients had MDS with increased blasts (MDS-IB1) and 10 (13.9%) had MDS-IB2. Regarding the classification risk based on the IPSS-R, 10 (13.89%) patients had very low risk IPSS-R, 18 (25.0%) had low risk, 12 (16.67%) had intermediate risk, 10 (13.89%) had high risk, and 9 (12.50%) had very high risk (Table 1).

Cytogenetic evaluation of bone marrow cells was performed for all Brazilian cases. Of those patients, 41/72 (56.9%) had a normal karyotype and 18/72 (25.0%) had an altered karyotype, of which 12 (16.7%) showed aneuploid karyotype, and 6 (8.3%) and 4 (5.6%) had alterations in chromosome five (-5/5q-) and chromosome seven (-7/7q-), respectively. In addition, 13 patients (18.1%) presented no metaphases (Table 1). The control group consisted of four elderly individuals without MDS or any other onco-hematological disease. Of these, three (75%) were female and one (25%) was male, with an average age of 76.5 years. All individuals in the control group showed normal results for cytogenetic analysis.

#### 2.2.2. The *USP15-USP7* Axis Is Downregulated in De Novo Brazilian MDS Patients

Gene expressions of *UBE2O*, *UBE2T*, *USP7*, and *USP15* were assessed in all MDS patients (*n* = 72) and controls (*n* = 4). Figure 3 displays the heterogeneity of expression of these genes in MDS patients.

We observed a significant decrease in *USP15* gene expression in MDS patients compared with the control group (*p* = 0.030) (Figure 4D). However, no significant associations were found in the expressions of the genes *UBE2T* (*p* = 0.152) (Figure 4A), *UBE2O* (*p* = 0.138) (Figure 4B), and *USP7* (*p* = 0.139) (Figure 4C) in our cohort. Nonetheless, Spearman’s correlation analysis (Figure 5A) revealed a strong, significant, and positive correlation between the *USP7* and *USP15* genes (r = 0.82; *p* < 0.0001) (Figure 5B), indicating that these genes work in a dependent manner, as a cascade of events in MDS. These results reinforce the importance of the *USP7* and *USP15* genes in MDS pathogenesis.

#### 2.2.3. Chromosomal Abnormalities Are Associated with *UBE2T* Upregulation in Brazilian MDS Patients

De novo MDS patients who presented chromosomal abnormalities exhibited significantly higher expression of the *UBE2T* gene (*p* = 0.0321) (Figure 6A) than de novo MDS patients without chromosomal abnormalities.

#### 2.2.4. *UBE2T* Downregulation Is Associated with Brazilian MDS Hypoplastic (MDS-h) Patients

Patients with Brazilian MDS-h exhibited significantly lower expressions of *UBE2T* (*p* = 0.033) compared with all other MDS clinical subgroups, particularly to MDS-RS-LB (Figure 7A).

## 3. Discussion

This report analyzes the expression of ubiquitination (*UBE2T* and *UBE2O*) and deubiquitination (*USP7* and *USP15*) genes in relation to pathogenesis and clinical variables in patients with MDS. Initially, our in-silico prediction results showed that *UBE2O*, *UBE2T*, and *USP7* were upregulated in MDS patients when compared with healthy individuals, but only in mononucleated cells collected from bone marrow samples. In contrast, our in vivo Brazilian validation cohort results demonstrated that only the *USP15* gene showed a downregulated expression compared with healthy individuals. Furthermore, the Spearman’s correlation analysis demonstrated a robust, significant, and positive correlation between the *USP7* and *USP15* genes, suggesting that these genes function in a dependent manner, acting as a cascade of events in MDS.

Ubiquitination is a post-translational enzymatic modification process in which proteins are tagged with ubiquitin molecules, marking them for degradation by the ubiquitin-proteasome system. This process is dynamic and can be reversed by deubiquitinases that remove ubiquitin molecules from their substrates, allowing the proteins to be rescued from proteasomal degradation [27]. Ubiquitination is responsible for regulating protein stability, and its dysregulation can contribute to the development of various human diseases, including cancer [28]. Similarly, dysregulation of deubiquitination processes can also lead to pathologies and diseases, such as hematologic malignancies [29].

One important gene involved in cellular deubiquitination mechanisms is *USP15*, which is involved in critical cellular and oncogenic processes, and its expression is deregulated in several types of cancers. Some authors have reported that the increased expression of *USP15* is related to the development of several types of cancers [30,31,32]. Specifically, for onco-hematological diseases, the *USP15* gene is usually highly expressed in human hematopoietic tissues and leukemias. Its depletion in murine progenitors and leukemia cells impairs cell expansion in vitro and increases genotoxic stress [33]. In leukemic cells, the *USP15* gene interacts with and stabilizes FUS (fused in sarcoma), a DNA repair factor, directly linking *USP15* to the DNA damage response, demonstrating the importance of DUBs in preserving normal hematopoiesis. Thus, *USP15* acts as a critical DUB in protecting the integrity of the genome in hematopoietic stem cells and leukemic cells. In AML, *USP15* is found upregulated compared with normal hematopoietic progenitor cells, and its inhibition, together with the following activation of *NRF2* (nuclear factor 2-related erythroid factor 2), leads to redox (oxidation-reduction) perturbations. On the other hand, in leukemic cells, USP15 is inessential for normal human and mouse hematopoietic cells in vitro and in vivo [25].

Furthermore, Niederkorn and collaborators [33] found that TIFAB (fork-associated B domain), a protein commonly involved in myeloid malignancies, regulates *USP15* signaling to substrates in hematopoietic cells. The expression of *TIFAB* in HSCs allows the signaling of *USP15* to different substrates, including *MDM2* and *KEAP1*, leading to a decrease in *TP53* expression, which may consequently promote leukemic transformation. In our study, we identified that the *USP15* gene was significantly reduced in MDS patients compared with healthy individuals (Figure 4D). It is important to emphasize that the *TIFAB* gene is located on chromosome five in the 5q31.1 region and is commonly deleted in hematological diseases, including MDS. It is also remarkable that only six (8.33%) patients were identified with the presence of chromosome five deletion, even if not isolated, indicating no prevalence of cases of 5q syndrome in the present study.

Understanding that the series of MDS patients evaluated in this study is not associated with the presence of a deletion in chromosome five, the *TIFAB* gene may be active and effectively recruiting *USP15* for the deubiquitination of its targets. However, when we identified that *USP15* had decreased expression in the cases evaluated, we hypothesized that this molecular event decreased *USP15* expression and maintained sufficient *TP53* expression to keep cell cycle control, preventing clonal evolution. Thus, we verified that *USP15* can be considered a possible new marker of genomic instability control for MDS. Future studies are needed to validate the role of *TIFAB* and its correlation with *USP15* in MDS.

Another deubiquitination enzyme is ubiquitin-specific protease 7 (*USP7*), an essential member of the USP DUB family that acts as a ~128 kDa cysteine protease with a significant role in genome stability. *USP7* regulates the p53/Mdm2 signaling axis [34]. Our in-silico results revealed that the *USP7* gene is upregulated in MDS patients. However, our in vivo analysis of bone marrow samples from Brazilian MDS patients showed that the *USP7* gene did not exhibit a differentiated expression profile compared with the control group and did not show any association with clinical variables or prognosis of the disease. Therefore, the impact of *USP7* on the pathogenesis of onco-hematological diseases, especially MDS, remains unclear. Regarding this, some studies suggest that increased *USP7* expression is associated with a variety of human malignancies, including prostate, breast, lung, cervical, and multiple myeloma cancers, where it regulates the activity of tumor promoter or suppressor proteins [33,35,36].

The role of USP7 is well-described in some types of onco-hematologic diseases. Agathangelou et al. (2017) reported an increase in *USP7* expression in Chronic Lymphocytic Leukemia (CLL) patients compared with healthy individuals. Loss or inhibition of *USP7* has also been shown to partially disrupt DNA repair by homologous recombination, leading to significant tumor cell death, independently of ATM and p53, through the accumulation of genotoxic levels of DNA damage [37]. Cartel et al. (2021) demonstrated that USP7 inhibition significantly reduces cell proliferation in vitro and in vivo, blocks the progression of DNA replication, and increases cell death in Acute Myeloid Leukemia (AML) [38]. However, in the case of MDS, only one study described *USP7* expression in a cohort of 28 Asian patients, which showed upregulation of *USP7* only in high-risk patients (SMD-EB-1 and MDS-EB-2). Notably, the authors did not evaluate the expression profile of *USP7* in patients with other clinical subtypes of the disease [39]. Therefore, our results demonstrate for the first time the activation of the *USP7-USP15* axis in MDS, indicating its role in controlling genomic instability in this disease. Finally, further studies are needed to elucidate the role of *USP7* in the pathogenesis of myelodysplastic syndrome.

After that, we identified that patients with MDS who had an altered karyotype had a significantly higher expression of *UBE2T* compared with those with a normal karyotype [5]. UBE2T belongs to the family of ubiquitin-conjugating enzymes, which were initially identified as essential elements in the Fanconi anemia (FA) pathway. In this pathway, UBE2T binds to FANCL, an E3 ubiquitin ligase, mediating the monoubiquitination of FANCD2, a protein that participates in DNA repair in the FA pathway [10,40]. The participation of UBE2T in FANCD2/FANCI monoubiquitination is a fundamental process for the repair of DNA damage, ensuring the maintenance of chromosomal stability. Failures in this process result in chromosomal aberrations, a known characteristic of Fanconi Anemia cells [41]. Furthermore, Ueki et al. [42] also reinforced the role of UBE2T in controlling the DNA repair mechanism when they reported that this target interacts with and co-localizes with the BRCA1/BRCA1-associated RING domain protein complex (BARD1) [42].

Our research group has been making efforts to assess the role of DNA repair genes in the pathobiology of Brazilian MDS. Based on gene expression analysis in CD34+ hematopoietic stem cells from Brazilian MDS patients, we identified that the *BRCA1* gene was downregulated in patients with >5% blast cells and in patients who died [43]. Interestingly, patients with altered karyotypes associated with increased *UBE2T* expression identified in the present study were also associated with increased bone marrow blasts. In addition, patients with decreased *BRCA1* expression (<0.009285) were associated with worse overall survival [43], reinforcing that the *BRCA1* gene is dysregulated in MDS. Therefore, we hypothesize that the increased expression of *UBE2T* in patients with altered karyotypes may be triggering the decrease in *BRCA1*, deactivating the homologous recombination mechanism that would reverse the cytogenetic alterations identified in these patients. Future studies are needed to validate this hypothesis and explore the correlation between *BRCA1* and *UBE2T*.

Finally, we evaluated the expression of *UBE2O*, a hybrid enzyme that possesses both E2 and E3 activities and can ubiquitinate proteins independently of E3 [6,11]. Some studies suggest that *UBE2O* has a significant role in several cancers. For example, in multiple myeloma (MM), UBE2O can lead c-Maf (a transcription factor expressed in MM) to degradation, inhibiting cell growth mediated by apoptosis and acting as a tumor suppressor [11]. However, in breast and prostate cancer, upregulation of *UBE2O* induces the proliferation of cancer cells, promoting the ubiquitination and degradation of AMPKα2 [17]. Additionally, patients with high *UBE2O* expression tend to have a higher risk of metastasis and poor prognosis in breast cancer [44]. Conversely, in mixed-lineage leukemia (MLL), depletion of *UBE2O* decreases the proliferation of leukemic cells by reducing the ubiquitination and degradation of the wild-type MLL protein, increasing its stability [18]. In our study, we did not identify any pathogenic or prognostic differences in *UBE2O* expression in MDS, probably due to the sample size evaluated in our Brazilian cohort. However, in silico analyses showed that *UBE2O* was downregulated in MDS cases. Notably, *UBE2O* is often amplified or mutated in multiple cancers, and its high expression is associated with a low survival rate in patients with gastric, lung, breast, and prostate cancer. Therefore, our results do not rule out the possibility that genomic alterations (e.g., mutations) in the *UBE2O* gene can be identified in MDS and be associated with the pathogenesis of the disease. Future studies are needed to confirm this hypothesis [6].

An important limitation of this study was the need to validate the impact of the identified profiles and gene expressions (*USP7*, *USP15*, *UBE2T*, and *UBE2O*) through both protein expression and Next-Generation Sequencing and mutation analysis, as these could affect the correct functioning of these genes in MDS patients.

## 4. Materials and Methods

### 4.1. In Silico Prediction of UBE2T, UBE2O, USP7, and USP15 Gene Expressions

To predict the expression profile of the *USP15*, *USP7*, *UBE2T*, and *UBE2O* genes in MDS, we used gene expression data based on analyses of Microarray databases from the Gene Expression Omnibus (GEO) database (https://www.ncbi.nlm.nih.gov/geo/) previously and recently published by Ribeiro-Junior et al. [26]. The authors performed a microarray re-analysis from primary mononucleated bone marrows cell (700 MDS patients and 47 healthy individuals) and CD34+ hematopoietic stem cell (392 MDS patients and 44 healthy individuals) samples from a total of 1092 MDS patients [26]. We then used R packages to compare the expression of the *USP15*, *USP7*, *UBE2T*, and *UBE2O* genes between the control and MDS groups using Student’s *t* test. We created boxplots using the “ggplot2” and “ggpubr” packages. A *p*-value of <0.05 was considered statistically significant.

### 4.2. Validation of UBE2T, UBE2O, USP7, and USP15 Gene Expressions in a Cohort of Brazilian MDS

#### 4.2.1. Patients and Sample Collection

Seventy-two patients with de novo MDS were diagnosed at the Federal University of Ceará (UFC)/Center for Research and Drug Development (NPDM) according to WHO 2022 criteria. The MDS patients were evaluated using the Revised International Prognostic Scoring System (IPSS-R). Four bone marrow samples were obtained from elderly healthy volunteers and used as control samples.

#### 4.2.2. Bone Marrow Aspiration and Processing

Bone marrow was obtained by aspiration from all patients by an experienced hematologist at Walter Cantídio University Hospital, Ceará, Brazil. The bone marrow samples were processed and analyzed at the Cancer Cytogenomics Laboratory (Federal University of Ceará) for cytogenetic and gene expression analysis.

#### 4.2.3. Cytogenetic Analysis by Conventional G-Band Karyotyping

Bone marrow cells were cultured for chromosomal analysis using conventional G-banding karyotype. Briefly, cultures were established in RPMI 1640 medium (Gibco, Grand Island, NY, USA) containing 30% fetal calf serum. After a 24 h cell culture, colcemid was added at a final concentration of 0.05 μg/mL to block the mitotic fuse. The cells were then harvested, exposed to a hypotonic solution (0.068 mol/L KCl), and fixed with Carnoy’s buffer (methanol/acetic acid, 1:3 proportion). The slides were prepared and stained using KaryoMAX^®^ Giemsa Stain. A minimum of 20 metaphases were analyzed, whenever possible, from each case, and chromosomes were analyzed with the CityVision Automated Karyotyping System (Applied Imaging, San Jose, CA, USA). This number of metaphases analyzed for each case (bone marrow samples, not peripheral blood) followed the recommendation from the International System for Human Cytogenomic Nomenclature—ISCN 2020 [45].

#### 4.2.4. Gene Expression Analysis

##### Total RNA Extraction and cDNA Synthesis

The extraction of total RNA from isolated primary mononuclear bone marrow (PMBM) cells was performed using TRizol Reagent™ (Invitrogen, Carlsbad, CA, USA), and cDNA synthesis was generated from the total RNA using a high-capacity cDNA Reverse Transcription kit^®^ (Applied Biosystems, San Jose, CA, USA), following the manufacturer’s protocol. The resulting cDNA samples were stored at −20 °C until further use.

##### Real-Time Quantitative PCR (qRT-PCR)

Reactions were performed on a 7500 Fast System^®^ (Applied Biosystems, Carlsbad, CA, USA) using TaqMan^TM^ assays (Applied Biosystems, Carlsbad, CA, USA). The TaqMan gene expression system was utilized to quantitatively measure the expression levels of *UBE2T* (Hs00928040_m1), *UBE2O* (Hs00222904_m1), *USP7* (Hs00931763_m1), and *USP15* (Hs00246242_m1).

To validate the stability of expression levels, Glyceraldehyde-3-Phosphate Dehydrogenase (GAPDH) and Hypoxanthine Phosphoribosyltransferase 1 (HPRT1) were chosen to normalize differences in input cDNA as endogenous controls; the RefFinder online bioinformatics tool (https://www.heartcure.com.au/reffinder/) was used. This tool integrates several computer programs, such as geNorm, Normfinder, BestKeeper, and the comparative Delta-Ct method, to compare and rank the candidate reference genes tested in the study. Each sample was analyzed in duplicate, and the expression ratios were calculated using the 2^−ΔCq^ method.

### 4.3. Statistical Analysis

Data on relative mRNA expression (∆Cq values—quantitative cycle) were expressed as the mean and range (maximum and minimum) to determine the possible association between gene expression and variables. Normality was evaluated using the Shapiro–Wilk test, and outliers were removed. Student’s *t*-test and one-way ANOVA with Tukey/Games Howell post hoc test were used when normality was detected. Homogeneity of variances was tested using Levene’s test. Pearson’s correlation test was used to obtain the r and r-square (r2) values. Plots for Pearson’s correlation analyses were created using the Morpheus online tool (https://software.broadinstitute.org/morpheus). Statistical analyses were performed using SPSS v. 21.0 (SPSS Inc., Chicago, IL, USA) and GraphPad Prism v. 6 (GraphPad Prism software, La Jolla, CA, USA). A probability level (*p*-value) of <0.05 was adopted.

## 5. Conclusions

In summary, we demonstrated for the first time that the *USP7*, *USP15*, and *UBE2T* genes might be related to MDS pathogenesis and prognosis, supporting the importance of these genes in understanding the etiology and prognostic stratification of Myelodysplastic Syndrome. Our results highlight that the differential gene expression of the *USP15-USP7* axis and *UBE2T* can play a crucial role in the control of genomic instability, which is one of the most significant features of MDS, particularly chromosomal abnormalities.

## Figures and Tables

**Figure 1 ijms-24-10058-f001:**
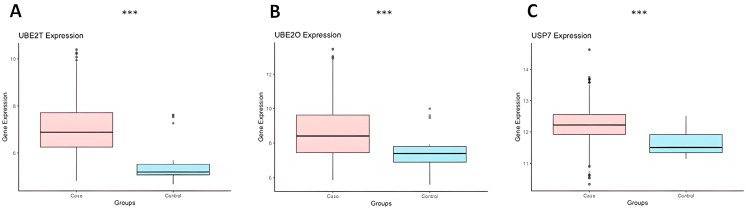
The expression of *UBE2T* (**A**), *UBE2O* (**B**), and *USP7* (**C**) genes in mononucleated cells collected from the bone marrow samples of MDS patients (*n* = 700) and controls (*n* = 47) based on in silico prediction microarray analysis. *** means *p*-value < 0.001.

**Figure 2 ijms-24-10058-f002:**
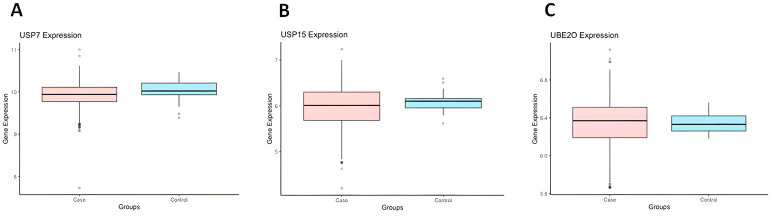
The expression of *USP7* (**A**), *USP15* (**B**), and *UBE2O* (**C**) genes in isolated CD34+ stem cells from the bone marrow samples of MDS patients (*n* = 700) and controls (*n* = 47) based on in silico prediction microarray analysis.

**Figure 3 ijms-24-10058-f003:**
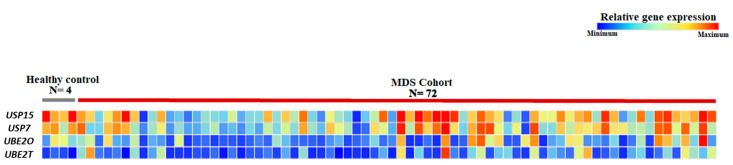
Heatmap of the relative expression levels of the *UBE2T*, *UBE2O*, *USP7*, and *USP15* genes in patients with MDS and healthy individuals (control group).

**Figure 4 ijms-24-10058-f004:**
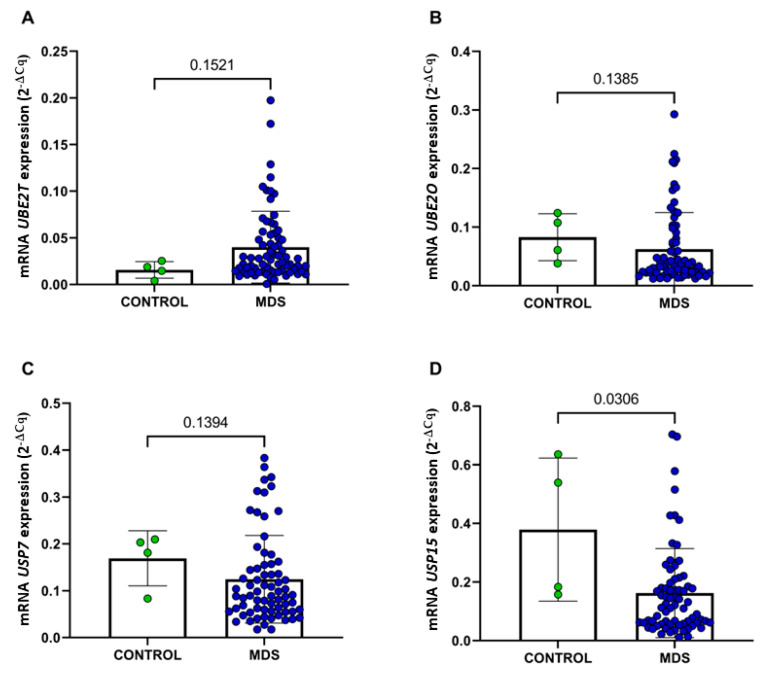
mRNA expression of the genes *UBE2T* (**A**), *UBE2O* (**B**), *USP7* (**C**), and *USP15* (**D**) in Brazilian MDS patients compared with controls. The *p*-value data are displayed in the center of the graph, above the connecting arrow between the graphs. Control, *n* = 4. MDS, *n* = 72.

**Figure 5 ijms-24-10058-f005:**
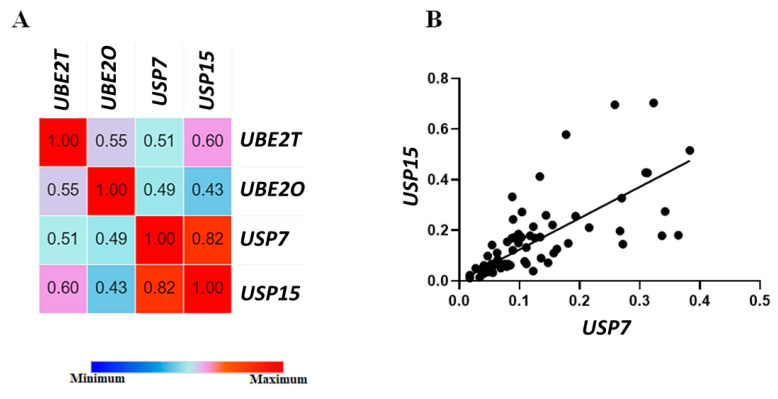
Spearman’s correlation analysis of *UBE2T*, *UBE2O*, *USP7*, and *USP15* genes in Brazilian MDS patients. (**A**) Representation of Spearman’s analyses of controls between the *UBE2T, UBE2O, USP7*, and *USP15* genes against their levels of gene expression in patients with MDS. (**B**) Spearman’s correlation plot between *USP7* and *USP15* genes in MDS patients.

**Figure 6 ijms-24-10058-f006:**
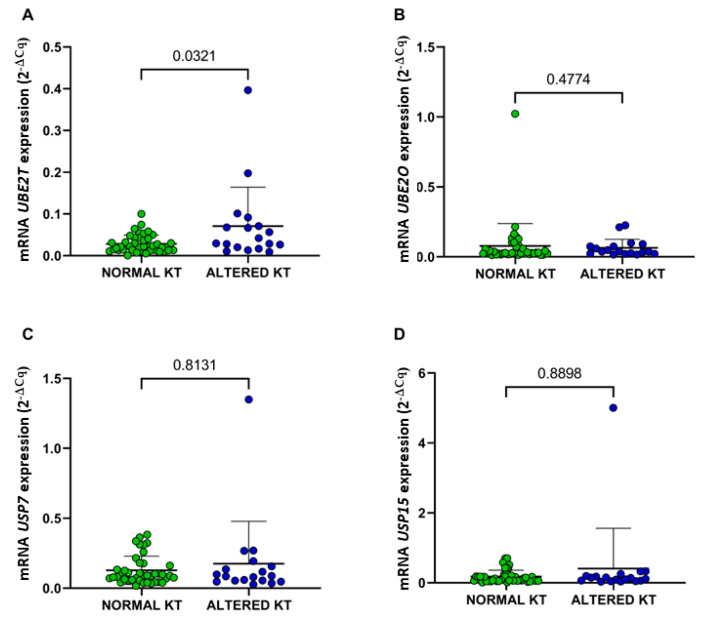
mRNA expression of the genes *UBE2T* (**A**), *UBE2O* (**B**), *USP7* (**C**), and *USP15* (**D**) according to the presence of chromosomal abnormalities in MDS patients. KT: Karyotype. Normal KT, *n* = 41. Altered KT, *n* = 18.

**Figure 7 ijms-24-10058-f007:**
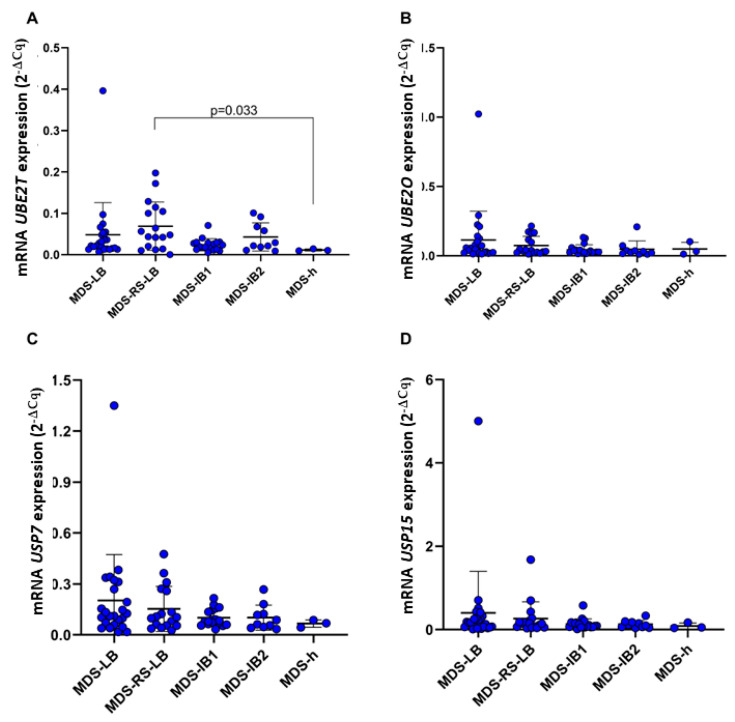
mRNA expression of the genes *UBE2T* (**A**), *UBE2O* (**B**), *USP7* (**C**), and *USP15* (**D**) according to MDS clinical classification based on WHO 2022. MDS-LB, *n* = 24. MDS-RS-LB, *n* = 17. MDS-IB1, *n* = 18. MDS-IB2, *n* = 10. MDS-h, *n* = 3.

**Table 1 ijms-24-10058-t001:** Cytogenetic characteristics and clinical classification of Brazilian patients with MDS.

Case	Age/ Gender	Karyotype Analysis (Based on ISCN)	WHO 2022	IPSS-R
1	62/M	47,XY,+8[6]/47,XY,del(7)(q32),+8[7]/46,XY[2]	MDS-IB2	Very high Risk
2	74/M	46,XY[11]	MDS-LB	Low risk
3	52/F	46,XX[8]	MDS-LB	Very low
4	82/M	No metaphase	MDS-RS-LB	-
5	66/F	46,XX[7]	MDS-LB	Low risk
6	81/F	46,XX[5]	MDS-RS-LB	Very low
7	47/M	47,XY,+mar[5]/46,XY,del(5)(q31)[5]/46,XY[15]	MDS-LB	Intermediate risk
8	81/M	46 XY[20]	MDS-LB	High risk
9	88/M	46 XY[6]	MDS-LB	Intermediate risk
10	71/M	46,XY[17]	MDS-RS-LB	Very low
11	68/M	No metaphase	MDS-IB2	-
12	79/M	46,XY[7]	MDS-LB	Very low
13	61/F	46,XX[20]	MDS-LB	Intermediate risk
14	82/M	No metaphase	MDS-LB	-
15	70/F	No metaphase	MDS-RS-LB	-
16	87/F	No metaphase	MDS-RS-LB	-
17	77/F	46,XX[27]	MDS-IB1	High risk
18	58/F	No metaphase	MDS-LB	-
19	84/F	No metaphase	MDS-RS-LB	-
20	72/F	No metaphase	MDS-IB2	-
21	71/F	46 XX[10]	MDS-RS-LB	Intermediate risk
22	77/F	No metaphase	MDS-LB	-
23	31/M	46,XY[24]	MDS-LB	Low risk
24	60/M	47,XY,+15[10]/46,XY[10]	MDS-RS-LB	Low risk
25	71/F	46,XX[5]	MDS-LB	Very low
26	43/M	46,XY[5]	MDS-LB	Low risk
27	60/F	No metaphase	MDS-LB	-
28	45/F	46,XX[20]	MDS-IB2	High risk
29	80/M	No metaphase	MDS-IB1	-
30	45/F	46,XX[7]	MDS-LB	Very low
31	79/M	46,XY,del(5)(q34),del(11)(q23)[7]/46,XY[5]	MDS-LB	High risk
32	76/F	46,XX[15]	MDS-RS-LB	Low risk
33	86/F	No metaphase	MDS-IB1	-
34	53/F	46 XX[20]	MDS-LB	Low risk
35	62/F	46 XX[10]	MDS hypoplastic	Low risk
36	74/M	No metaphase	MDS-RS-LB	-
37	81/M	46,XY[18]	MDS hypoplastic	Low risk
38	86/F	46,XX[20]	MDS-RS-LB	Intermediate risk
39	58/M	46,XY,del(5)(q32)[3]/46,XY,del(5)(q32),del(7) (q36)[3]/46,XY,5,+mar[9]/46,XY[7]	MDS-IB1	Very high risk
40	92/M	46,XY,del(5)(q32)[13]/46,XY[17]	MDS-RS-LB	Low risk
41	48/F	46,XY[4]	MDS-LB	Very low
42	71/M	46,XY[20]	MDS-IB1	High risk
43	76/F	46,XX[20]	MDS-RS-LB	Low risk
44	75/M	92, XXYY<4n>[4]/46,XY[8]	MDS-IB1	Very high risk
45	82/M	92,XXYY<4n>[5]/46,XY[15]	MDS-LB	Intermediate risk
46	77/F	46,XX[20]	MDS-IB1	Low risk
47	63/F	92,XXXX<4n>[3]/46,XX[17]	MDS-LB	Low risk
48	79/M	46,XY[20]	MDS-IB1	Intermediate risk
49	82/F	47,XX,+8[9]/47,XX,+8,del(20)(q12)[5]/46,XX[6]	MDS-RS-LB	Intermediate risk
50	45/F	47,XX,+6[3]/46,XX[17]	MDS-LB	Low risk
51	43/F	46,XX[20]	MDS-IB1	Intermediate risk
52	79/F	46,XX[12]	MDS-IB2	High risk
53	81/F	46,XY[20]	MDS-IB2	High risk
54	55/M	45,XY,-7[15]/46,XY,-7,+mar[5]	MDS-IB2	Very high risk
55	89/M	46,XY,t[5;19)(q13.2;q13.4)[3]/46,XY,t[5;19)(q13.2;q13.4),t(8,21)(q21.3;q22.12)[3]/46,Xydel(X)(q21),t(5;19)(q13.2;q13.4),t(8;21)(q21.3;q22.12)[5]/46,XY[9]	MDS-IB2	Very high risk
56	46/F	46,XX[20]	MDS-LB	Low risk
57	93/F	46,XX,+8[12]/46,XX[8]	MDS-IB1	Very high risk
58	84/M	46,XY[20]	MDS-IB1	High risk
59	80/F	46,XY[20]	MDS-RS-LB	Low risk
60	87/M	46,XY[20]	MDS-LB	Very low
61	73/F	46, XX[20]	MDS hypoplastic	Very low
62	71/M	46,XY[15]	MDS-RS-LB	Very low
63	76/M	46, XY, del (5)(q31)[2]/46,XY, del(5)(q31)-7,+8[16]	MDS-IB2	Very high risk
64	58/M	46,XX[20]	MDS-IB1	Intermediate risk
65	84/F	46,XX[20]	MDS-IB1	High risk
66	72/F	46,XX,inv(3)(q21q26.2)[16]/46,XX[4]	MDS-IB1	Very high risk
67	71/F	46,XX[20]	MDS-IB1	Low risk
68	72/M	46,XY[20]	MDS-IB1	Intermediate risk
69	91/F	46,XX[20]	MDS-IB1	High risk
70	64/M	46,XY[20]	MDS-IB1	Intermediate risk
71	87/F	46, XX, del(11)(q21)[16]/46,XX[4]	MDS-RS-LB	Low risk
72	86/F	46,XX,del(5)(q14)[5]/46,XX[15]	MDS-IB2	Very high risk

Legend. M. Male; F. Female; “-“. Absence of IPSSR classification; MDS. Myelodysplastic neoplasm; IB. MDS with increased blasts; RS. Ring sideroblasts; LB. MDS with low blast; WHO. World Health Organization Classification of Hematolymphoid Tumors; IPSS-R. Revised International Prognostic Score System; ISCN. International System for Human Cytogenomic Nomenclature.

## Data Availability

Not applicable.

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
