# Peer review of "USP15-USP7 Axis and UBE2T Differential Expression May Predict Pathogenesis and Poor Prognosis in De Novo Myelodysplastic Neoplasm"

_ijms, 2023, doi:10.3390/ijms241210058_

Round 1

Reviewer 1 Report

The authors of this manuscript conducted an analysis of the expression of four genes in MDS patients using data from both the GEO databases and patient samples. They also examined the correlation between gene expression and chromosomal abnormalities and clinical classification of MDS. Typically, large amounts of data from GEO are used to support conclusions drawn from patient samples or biochemical experiments. However, in this manuscript, the data from GEO and Brazilian patients contradicted each other. Additionally, USP15, the only gene whose expression was downregulated in Brazilian patients, was not associated with chromatin abnormalities or clinical classification of MDS. As a result, the data presented in the manuscript are inconsistent and insufficient to support the authors' claims.

The authors have listed Ubiquitination and Deubiquitination as keywords and stated that their aim is “to evaluate the expression of these genes to identify potential targets of ubiquitination and deubiquitination in the pathobiology of de novo MDS”. However, it is important to note that Ubiquitination is a protein post-translational modification. Therefore, it is not possible for the authors to directly evaluate the level of ubiquitination in vivo or identify targets of ubiquitination solely through gene expression analysis.

The authors have chosen two Ubiquitin-conjugating enzymes and two Deubiquitinating enzymes that are involved in DNA damage repair for their study. However, it is unclear whether there is an association between these four genes, as this is not stated in the paper. Furthermore, these genes exhibit different expression patterns in MDS patients. Therefore, it is unclear why the authors have chosen to study all four genes simultaneously.

Fig 1-2 is missing the expression levels of USP15.

Fig 3-7, it is unclear which cells the authors used to detect gene expressions.

Fig 6-7, the authors should include the exact value of n in the figure legends.

In line 177, the authors state that their results reinforce the importance of the USP7 and USP15 genes in the pathogenesis of MDS. However, it is unclear whether there are any survival data or other evidence demonstrating the clinical significance of abnormal expression of these genes in MDS.

Line 264, is it correct to refer to the new clinical classification established by the WHO in 2016 in this context?

In lines 297 and 352-354, the authors suggest that USP15 and USP7 may play a crucial role in the control of genomic instability in MDS, and that USP15 may be a possible new marker for this process. However, these claims not be supported by the data presented in Fig 6-7, which show no significant changes in the expressions of USP15 and USP7.

The Results section of the paper is too concise and lacks sufficient explanation for the observed findings. For example, in section 2.1, the authors report upregulation of UBE2T, UBE2O, and USP7 in mononuclear BM cells, but not in CD34+ stem cells of MDS patients. However, the authors have not provided an explanation for why this difference in expression is observed. In section 2.2.4, the authors report that downregulation of UBE2T is associated with MDS-h patients, but it is not clear what this association means.

In the Discussion section, the authors make too much assumptions without providing adequate evidence in this manuscript. 

Author Response

Reviewer's 1 comment: The authors of this manuscript conducted an analysis of the expression of four genes in MDS patients using data from both the GEO databases and patient samples. They also examined the correlation between gene expression and chromosomal abnormalities and clinical classification of MDS. Typically, large amounts of data from GEO are used to support conclusions drawn from patient samples or biochemical experiments. However, in this manuscript, the data from GEO and Brazilian patients contradicted each other.

Author’s reply: In silico prediction data from GEO were used to predict the expression profiles of UBE2T, UBE2O, USP7 and USP15 in the pathogenesis of MDS. However, these studies are usually performed on samples from European Caucasian patients. Several studies report on clinical differences in MDS imposed by race. Thus, our in-silico prediction results guided the validation of these results in a Brazilian cohort. Finally, the in silico results do not contradict each other, but are validated in another cohort (Brazilian cohort).

Reviewer's 1 comment: Additionally, USP15, the only gene whose expression was downregulated in Brazilian patients, was not associated with chromatin abnormalities or clinical classification of MDS. As a result, the data presented in the manuscript are inconsistent and insufficient to support the authors' claims.

In lines 297 and 352-354, the authors suggest that USP15 and USP7 may play a crucial role in the control of genomic instability in MDS, and that USP15 may be a possible new marker for this process. However, these claims not be supported by the data presented in Fig 6-7, which show no significant changes in the expressions of USP15 and USP7.

Author’s reply: In our in-silico prediction analysis, we did not identify USP15 expression results in MDS. However, our validation results showed a decrease in the expression of USP15 in MDS when compared to healthy individuals, demonstrating its potential role in the pathogenicity of MDS, even without clinical or prognostic associations in the disease. In fact, USP7 was not significantly associated with MDS in the case x control evaluation, probably due to the evaluated sample size; however, its strong correlation with USP15 reinforces its activation in disease.

Reviewer's 1 comment: The authors have listed Ubiquitination and Deubiquitination as keywords and stated that their aim is “to evaluate the expression of these genes to identify potential targets of ubiquitination and deubiquitination in the pathobiology of de novo MDS”. However, it is important to note that Ubiquitination is a protein post-translational modification. Therefore, it is not possible for the authors to directly evaluate the level of ubiquitination in vivo or identify targets of ubiquitination solely through gene expression analysis.

Author’s reply: Gene expression studies are extremely important for the development of new biomarkers for disease pathogenicity. This is because gene expression is the process by which genetic information is used to produce proteins and other important molecules. Therefore, the analysis of gene expression can provide valuable information about which genes are active in a given condition, as well as the intensity of the activity of these genes. However, we understand that our study has the limitation of not having evaluated the protein expression profile of UBE2T, UBE2O, USP7 and USP15 in MDS. This point of limitation has been added to the penultimate paragraph of the discussion.

Reviewer's 1 comment: The authors have chosen two Ubiquitin-conjugating enzymes and two Deubiquitinating enzymes that are involved in DNA damage repair for their study. However, it is unclear whether there is an association between these four genes, as this is not stated in the paper.

Author’s reply: Our results demonstrated that only the USP7 and USP15 genes are correlated in MDS, probably acting in the genomic instability process of the disease.

Reviewer's 1 comment: Furthermore, these genes exhibit different expression patterns in MDS patients. Therefore, it is unclear why the authors have chosen to study all four genes simultaneously.

Author’s reply: To date, there are no data associating the expression of ubiquitination and deubiquitination markers as pathogenic in MDS or associated with clinical or prognostic variables of the disease. The choice to evaluate the USP7, USP15, UBE2T and UBE2O genes in this study was because these targets have been associated with other types of neoplasms, hematological or not, and are associated with DNA damage, an important marker for MDS.

Reviewer's 1 comment: Fig 1-2 is missing the expression levels of USP15.

Author’s reply: Microarray panels from GEO, both for primary bone marrow cells and for CD34+ stem cells, did not show levels of USP15 expression in MDS. Therefore, no results could be presented.

Reviewer's 1 comment: Fig 3-7, it is unclear which cells the authors used to detect gene expressions.

Author’s reply: All of our validation results for the expression of the USP7, USP15, UBE2T and UBE2O genes in the Brazilian cohort of patients with MDS were obtained from samples of primary bone marrow cells, as presented in items 2.2.1 and 4.2.4.1.

Reviewer's 1 comment: Fig 6-7, the authors should include the exact value of n in the figure legends.

Author’s reply: Gene expression analyzes were effectively performed in all 72 Brazilian patients diagnosed with MDS and in the 4 healthy individuals included as a control group. Descriptions of the clinical characteristics of the patients are described in Table 1.

Reviewer's 1 comment: In line 177, the authors state that their results reinforce the importance of the USP7 and USP15 genes in the pathogenesis of MDS. However, it is unclear whether there are any survival data or other evidence demonstrating the clinical significance of abnormal expression of these genes in MDS.

Author’s reply: In the present study, the impact of USP7, USP15, UBE2T and UBE2O gene expressions on MDS patient survival was not investigated. Nonetheless, case-control studies on gene expression can play a crucial role in the discovery of new pathogenic disease biomarkers. These studies compare the gene expression patterns of individuals who have a particular disease (the cases) with those who do not (the controls). By analyzing the differences in gene expression between the two groups, researchers can identify genes that are upregulated or downregulated in the cases, and that may be involved in the development or progression of the disease. Identifying such genes can help researchers develop new biomarkers for the disease, which can be used for early detection, diagnosis, and monitoring of the disease.

Reviewer's 1 comment: Line 264, is it correct to refer to the new clinical classification established by the WHO in 2016 in this context?

Author’s reply: The text has been corrected.

Reviewer 2 Report

The Authors aimed to evaluate the expression of USP7, USP15, UBE2O, and UBE2T genes in Myelodysplastic neoplasm (MDS) to identify possible targets of ubiquitination and deubiq-24 uitination in MDS pathobiology. 

The topic is interesting and the study well designed.

Introduction ok.

Methods: clearly described. The Authors performed expression analysis for UBE2O, UBE2T, USP7, and USP15 was based on microarray data from 1092 patients. 

However, they stated that analysis were made on MDS patients (n=700) and controls (n=47). Please check about numbers and specify further. Now, it might be confusing.

Is there any difference in expression between different stages/diseases of MDS?

Please discuss further on the possible therapeutic effects of these findings.

Author Response

Reviewer's 2 comment: The Authors aimed to evaluate the expression of USP7, USP15, UBE2O, and UBE2T genes in Myelodysplastic neoplasm (MDS) to identify possible targets of ubiquitination and deubiq-24 uitination in MDS pathobiology. 

The topic is interesting and the study well designed. Introduction ok. Methods: clearly described. The Authors performed expression analysis for UBE2O, UBE2T, USP7, and USP15 was based on microarray data from 1092 patients. However, they stated that analysis were made on MDS patients (n=700) and controls (n=47). Please check about numbers and specify further. Now, it might be confusing. Is there any difference in expression between different stages/diseases of MDS?

Author’s reply: Topic 4.1 on methods has been modified.

Reviewer's 2 comment: Please discuss further on the possible therapeutic effects of these findings.

Author’s reply: Our results reinforce the role of USP7, USP15 and UBE2T genes in the pathobiology of MDS. Future studies are needed to demonstrate the impact of these genes on therapeutic possibilities for MDS.

Reviewer 3 Report

 Reviewer comments and suggestions

 This study aimed to evaluate the expression of USP7, USP15, UBE2O, and UBE2T genes in Myelodysplastic neoplasm (MDS) to recognize possible targets of ubiquitination and deubiquitination in MDS pathobiology. For this study, the authors used eight datasets from the Gene Expression Omnibus (GEO) database analyzed in 1092 MDS patients and healthy controls. The results reported in the study showed that UBE2O, UBE2T, and USP7 were upregulated in MDS patients compared to healthy individuals. In contrast, only the USP15 gene showed a down-regulated expression compared to healthy individuals (p=0.03). Additionally, the upregulation of UBE2T expression was recognized in MDS patients with chromosomal abnormalities. Finally, the study concluded that USP7 and USP15 genes were strongly correlated with MDS (r=0.82; r2=0.67; p<0.0001). 

Overall, the manuscript was well written. However, a few concerns/comments needed to be explained/modified. 

  1. Line 33, please mention MDS hypoplastic instead of MDS-h
  2. Lines 50-51 need specific references.
  3. Line 97 Please explore the term.
  4. Line 221-222 is this sufficient to cover this section, please explore it well.
  5. Discussion first para The first para needs the novelty of the study, these points were already mentioned in the previous sections.
  6. Line 242 “An in-silico analysis revealed that the USP7 gene is upregulated in MDS patients”. Please mention the study 
  7. I observed that USP7 gene and other genes the authors mention in the MS some time italic and sometime non-italic please be consistent. 
  8. Line 280-284 It is important for the authors to mention the table or figures that they discuss to easily follow up on your paper.
  9. Line 310-311 please explain it well.
  10. Line 337 why, please mention the possible reason for this.
  11. All references need to be modified based on MDPI GUIDELINES

Author Response

Reviewer's 3 comment: This study aimed to evaluate the expression of USP7, USP15, UBE2O, and UBE2T genes in Myelodysplastic neoplasm (MDS) to recognize possible targets of ubiquitination and deubiquitination in MDS pathobiology. For this study, the authors used eight datasets from the Gene Expression Omnibus (GEO) database analyzed in 1092 MDS patients and healthy controls. The results reported in the study showed that UBE2O, UBE2T, and USP7 were upregulated in MDS patients compared to healthy individuals. In contrast, only the USP15 gene showed a down-regulated expression compared to healthy individuals (p=0.03). Additionally, the upregulation of UBE2T expression was recognized in MDS patients with chromosomal abnormalities. Finally, the study concluded that USP7 and USP15 genes were strongly correlated with MDS (r=0.82; r2=0.67; p<0.0001). Overall, the manuscript was well written. However, a few concerns/comments needed to be explained/modified. 

Line 33, please mention MDS hypoplastic instead of MDS-h

Author’s reply: The text has been corrected.

Lines 50-51 need specific references.

Author’s reply: References have been included.

Line 97 Please explore the term.

Author’s reply: The authors did not explore the potential therapeutic impacts of USP7 in different pathologies, as shown by the study by Wang et al. 2019 [21], as it was not the scope of the study. We have included this excerpt from the introduction as it reinforces the importance of studying USP7 as a possible marker of pathogenicity in MDS.

Line 221-222 is this sufficient to cover this section, please explore it well.

Author’s reply: The text has been corrected.

Discussion first para The first para needs the novelty of the study, these points were already mentioned in the previous sections.

Author’s reply: The discussion section was rewritten in order to adapt to the reviewer's recommendation.

Line 242 “An in-silico analysis revealed that the USP7 gene is upregulated in MDS patients”. Please mention the study 

Author’s reply: The text has been corrected.

I observed that USP7 gene and other genes the authors mention in the MS some time italic and sometime non-italic please be consistent. 

Author’s reply: We revised the text and corrected the names of the genes that were not in italics.

Line 280-284 It is important for the authors to mention the table or figures that they discuss to easily follow up on your paper.

Author’s reply: We identified the figure correctly in the text.

Line 310-311 please explain it well.

Author’s reply:  The study by Ueki et al [42] elucidates the role of the UBE2T gene on the DNA repair mechanism on its interaction with BRCA1. This reference links with previous results from our research group, which reinforces the impact of the BCRA1 gene on the process of genomic instability in MDS.

Line 337 why, please mention the possible reason for this.

Author’s reply: We added a justification to the text.

All references need to be modified based on MDPI GUIDELINES

Author’s reply: The list of references has been adapted to the format of the MDPI guidelines.

Round 2

Reviewer 1 Report

The revised manuscript has significantly improved its narrative rigor. However, I still believe that the Results section requires further elaboration to adequately explain the observed findings.

As the expression of USP15 was not detected in the GEO analysis, it should be removed from the Figure 1 legend.

Fig 6-7, the authors should include the specific value of n for each group.

Author Response

The revised manuscript has significantly improved its narrative rigor. However, I still believe that the Results section requires further elaboration to adequately explain the observed findings.

Review's 1 comment: As the expression of USP15 was not detected in the GEO analysis, it should be removed from the Figure 1 legend.

Author's reply: We appreciate the reviewer's comment. We inform you that the legend of figure 1 has been modified.

Review's 1 comment: Fig 6-7, the authors should include the specific value of n for each group.

Author's reply:  We appreciate the reviewer's comment. We inform you that we have added the numbers of samples, by group, in the figures of the article.